# Isolation of a Human Betaretrovirus from Patients with Primary Biliary Cholangitis

**DOI:** 10.3390/v14050886

**Published:** 2022-04-24

**Authors:** Mariam Goubran, Weiwei Wang, Stanislav Indik, Alexander Faschinger, Shawn T. Wasilenko, Jasper Bintner, Eric J. Carpenter, Guangzhi Zhang, Paulo Nuin, Georgina Macintyre, Gane K.-S. Wong, Andrew L. Mason

**Affiliations:** 1Center of Excellence for Gastrointestinal Inflammation and Immunity Research, University of Alberta, Edmonton, AB T6G 2E1, Canada; mgoubran@ualberta.ca (M.G.); wangweiwei301@gmail.com (W.W.); shawn.wasilenko@ualberta.ca (S.T.W.); j.bintner@gmail.com (J.B.); guzzhca@yahoo.com (G.Z.); georgina.macintyre@ualberta.ca (G.M.); gane@ualberta.ca (G.K.-S.W.); 2Department of Virology, University of Veterinary Medicine, A-1210 Vienna, Austria; stanislav.indik@vetmeduni.ac.at (S.I.); alexander.faschinger@gmx.at (A.F.); 3Department of Biological Sciences, University of Alberta, Edmonton, AB T6G 2E9, Canada; ejc@ualberta.ca; 4Department of Medical Genetics, University of Alberta, Edmonton, AB T6G 2E1, Canada; nuinsuan@ualberta.ca; 5Li Ka Shing Institute of Virology, University of Alberta, Edmonton, AB T6G 2E1, Canada

**Keywords:** biliary epithelial cells (BEC), common insertion sites (CIS), human betaretrovirus (HBRV), mouse mammary tumor virus (MMTV), primary biliary cholangitis (PBC)

## Abstract

A human betaretrovirus (HBRV) has been linked with the autoimmune liver disease, primary biliary cholangitis (PBC), and various cancers, including breast cancer and lymphoma. HBRV is closely related to the mouse mammary tumor virus, and represents the only exogenous betaretrovirus characterized in humans to date. Evidence of infection in patients with PBC has been demonstrated through the identification of proviral integration sites in lymphoid tissue, the major reservoir of infection, as well as biliary epithelium, which is the site of the disease process. Accordingly, we tested the hypothesis that patients with PBC harbor a transmissible betaretrovirus by co-cultivation of PBC patients’ lymph node homogenates with the HS578T breast cancer line. Because of the low level of HBRV replication, betaretrovirus producing cells were subcloned to optimize viral isolation and production. Evidence of infection was provided by electron microscopy, RT-PCR, in situ hybridization, cloning of the HBRV proviral genome and demonstration of more than 3400 integration sites. Further evidence of viral transmissibility was demonstrated by infection of biliary epithelial cells. While HBRV did not show a preference for integration proximal to specific genomic features, analyses of common insertion sites revealed evidence of integration proximal to cancer associated genes. These studies demonstrate the isolation of HBRV with features similar to mouse mammary tumor virus and confirm that patients with PBC display evidence of a transmissible viral infection.

## 1. Introduction

### 1.1. Human Betaretrovirus

A human betaretrovirus (HBRV, previously human mammary tumor virus) has been linked with breast cancer, lymphoma and female hormone responsive tumors [1,2,3,4,5,6,7,8,9,10,11,12,13,14,15,16,17,18]. The same agent has been characterized in patients with the autoimmune liver disease, primary biliary cholangitis (PBC, formerly primary biliary cirrhosis), and detected in the liver of patients with autoimmune hepatitis, alcohol use disorder, and hepatocellular carcinoma [19,20,21,22,23]. The closely related betaretrovirus, mouse mammary tumor virus (MMTV), is the causal agent of breast cancer, lymphoma and renal cancer in mice [16,24]. MMTV has also been linked with the development of cholangitis in the NOD.c3c4 autoimmune biliary disease mouse model of PBC [25,26].

HBRV is genetically indistinguishable from MMTV [20] and more closely related to the mammalian exogenous and endogenous retroviruses, such as Mason-Pfizer monkey virus and the Jaagsiekte sheep retrovirus, than the human endogenous HERV-K elements that form a separate branch on the betaretrovirus phylogenetic tree [27]. However, the nomenclature of the endogenous betaretroviruses HERV-K family can be confusing since they are referred to as human endogenous MMTV-like (HML) elements. Nevertheless, HBRV is the only exogenous betaretrovirus characterized in humans, as none of the HERV-K elements have been characterized as exogenous agents.

MMTV is both an exogenous and endogenous betaretrovirus that can be transmitted vertically as an endogenously expressed viral particle or exogenously passaged in breast milk [23]. Whereas, HBRV is not encoded in the human genome (albeit mistakenly reported as such [28,29]). The mode of HBRV transmission remains unknown. Similar to MMTV, betaretrovirus particles were detected by electron microscopy in milk from breast cancer patients and HBRV has been detected in saliva [17,30]. It is probable that HBRV was initially transmitted as a zoonosis from mice, and because HBRV has been found in the dental callus from skulls dating back to the copper age, this may have coincided with the development of agriculture [31].

The potential role of MMTV as a zoonosis has been questioned due to the concern that the human orthologue of the murine entry receptor, transferrin receptor 1, is insufficient to engage virions [32,33]. However, MMTV has subsequently been shown to infect several human cell types, including HEK 293, HeLa cells and Hs578T mammary gland cells, as demonstrated by the detection of provirus integration sites and inhibition of infection by neutralizing anti-MMTV serum [34,35,36].

### 1.2. Primary Biliary Cholangitis

Primary biliary cholangitis is an autoimmune liver disease characterized by progressive immune destruction of intrahepatic bile ducts and production of anti-mitochondrial antibodies (AMA) [37]. PBC predominantly occurs in women and is thought to occur as a result of an environmental trigger in a genetically susceptible host. Many infectious agents as well as xenobiotics have been proposed as etiological agents for PBC but no causal link has been firmly established to date [23]. To investigate a potential infectious etiology of PBC, we conducted representational difference analyses to uncover retroviral sequences in patient liver samples, Western blot studies that showed reactivity to retroviral antigens and electron microscopy studies that uncovered virus-like particles in biliary epithelium [21,38,39]. We subsequently cloned nucleic acid sequences from perihepatic lymph nodes and biliary epithelial cells with a variable 93% to 97% nucleotide sequence identity with MMTV [20,21]. Researchers unable to detect evidence of HBRV in liver disease patients suggested that the final evidence for a role of betaretrovirus in PBC could be provided by the direct demonstration of proviral integrations [40,41]. Therefore, we investigated the frequency of HBRV proviral integrations in patients undergoing liver transplantation using ligation mediated (LM)-PCR and next generation sequencing (NGS). We evaluated biliary epithelium extracted ex vivo, perihepatic lymph nodes and liver DNA and identified more than 1500 HBRV proviral integrations in the majority of PBC patients studied and in patients with autoimmune hepatitis (AIH), a related liver disease [19].

In PBC patients, the recorded prevalence of infection varied with the samples studied and methods employed to detect HBRV: (i) in perihepatic lymph nodes, the frequency of detection was 75% by RT-PCR and immunochemistry, and 45% for proviral integrations detected by LM-PCR and NGS; (ii) in biliary epithelial cells ex vivo, the frequency of infection was 75% by in situ hybridization, whereas 58% of samples were positive for proviral integrations by LM-PCR and NGS; and (iii) in the liver, we discovered HBRV RNA in 29% of PBC patients by RT-PCR, proviral integrations in 13%, and HBRV DNA in 17% of patients detected by nested PCR [19,21]. In contrast, immunoreactivity was observed in (iv) 50% of PBC patients’ peripheral blood mononuclear cells using an interferon-γ release assay [42], and (v) 11.5% demonstrated seroreactivity using an in house ELISA assay with HBRV Env gp52 protein expressed in HEK 293 cells [43]. The role of HBRV in autoimmune liver disease remains controversial because reproducible diagnostic assays are required to conduct confirmatory epidemiological studies.

### 1.3. Linking HBRV with PBC

The link of HBRV infection and PBC has been strengthened in clinical trials showing that combination antiretroviral therapy positively impacts on the biochemical and histological disease process [44,45,46]. However, there are no robust methods to prove that a relatively common infectious agent causes a rare and chronic disease that only occurs on a specific genetic background. An approach we initially employed was the demonstration of Koch’s postulates in vitro. A specific phenotypic appearance has been described in the cholangiocytes of PBC patients, and monocytes in perihepatic lymph nodes that demonstrate both an increased and aberrant expression of mitochondrial autoantigens [47]. The over-production of usually sequestered proteins is thought to lead to loss of tolerance to self and generation of anti-mitochondrial antibodies. Therefore, we tested the hypothesis that this disease-specific phenotype was triggered by HBRV infection and eventually demonstrated that even pure MMTV virions could induce this phenotype [21,48].

To provide further evidence that HBRV is a human pathogen, we now describe the isolation of HBRV using co-cultivation of Hs578T human breast cancer cells and lymph node homogenates from patients with PBC. We chose Hs578T cells because they are permissive for MMTV infection, they exhibit no evidence of proviral MMTV-like sequences in their genomic DNA, and the cells have not been exposed to murine retroviruses nor passaged through mice [34,36,49,50,51]. Characterization of the viral particles derived from co-cultured PBC conditioned media revealed a betaretrovirus morphology, and the proviral genome shared close identity with HBRV and MMTV nucleic acid sequences [18,20]. The HBRV isolates were passaged in culture and shown to infect human primary cholangiocytes. The evaluation of common integration sites (CIS) derived from over 5000 in vitro and in vivo insertions revealed that HBRV may insert proximal to cancer associated genes, comparable with MMTV [52].

## 2. Materials and Methods

### 2.1. Co-Cultivation Studies

Perihepatic lymph nodes were obtained at the time of liver transplantation from four patients with PBC diagnosed by standard criteria [37], two patients with cryptogenic cirrhosis and one patient with erythropoietic protoporphyria and stored at −80 °C. The lymph node homogenates for co-culture studies were processed by grinding frozen tissue with a pestle and mortar in liquid N2, resuspending the powder in phosphate buffered saline (0.1 g/mL), which was further homogenized using a Dounce tissue grinder. The supernatants were clarified by centrifuge for 5 min at 3500× *g* and polybrene as added to a final concentration of 8 μg/mL. Hs578T cells were obtained from ATCC (HTB-126) and grown on 6-well plates to 30% confluence in Dulbecco’s modified Eagle’s medium supplemented with 10% FBS in humidified air containing 5% CO_2_ at 37 °C. Co-cultures were performed in triplicate using inserts containing either 1 mL of lymph node homogenate and 10 nM dexamethasone or PBS control. Following 24-h incubation, the homogenates were removed and replaced with fresh medium containing 10 nM dexamethasone. After 48 h, the cells were expanded in T25 flasks and supernatant was collected after another 96 h for detection of HBRV. Hs578T cells with evidence of HBRV infection were isolated by three rounds of limiting dilution in media containing 10 nM dexamethasone (Figure 1).

For passage of infectious virus particles, pooled supernatants were filtered and used in co-culture with biliary epithelial cells extracted from explants derived from liver transplant recipients without biliary disease. This extraction methodology uses a standard procedure that typically isolates the smaller biliary epithelial cells affected in patients with PBC [19,21]. The biliary epithelial cells were initially derived by mechanical dissection of the liver explant, followed by a density gradient centrifugation and then immuno-capture with anti-CD326 (EpCam) microbeads (Miltenyi Biotec, Bergisch Gladbach, Germany), prior to maintenance and passage in biliary epithelial cell growth media [53].

### 2.2. Detection of HBRV

All experiments were performed with privileged cell culture equipment and extraction rooms for human cells without exposure to the Mm5MT MMTV producing cell line. 

Reverse transcriptase (RT) activity was measured in supernatants using an HS-Mg RT Activity Kit following manufacturer’s instructions (CavidiTec, Uppsala, Sweden) [54]. 

Quantitative (q)RT-PCR studies to measure HBRV RNA were performed using RNA extracted from supernatant or total RNA extracted from 1 × 10^6^ Hs578T cells using approximately a tenth of the cDNA. The assay was conducted using a Taqman 7300 Real Time PCR system with primers complementary to HBRV *pol* or *env* genes using serial dilutions of RSV:MMTV plasmid to calibrate sensitivity, as previously described [54].

In situ hybridization was performed to detect HBRV RNA (ViewRNA assay) in cell culture using probes designed and synthesized by Panomics from highly conserved *gag-pro-pol* genes, as described [19]. Between 10^3^ to 10^4^ cells were loaded in duplicates into a 96-well assay plate, fixed with 4% formaldehyde, digested with Proteinase K and hybridized in solution containing the probe set for 3 h. Cells were further stained with DAPI and viewed using fluorescent microscopy (Observer z1, ZEISS). Images were acquired with an AxioVision (ZEISS) microscope from 3 different areas using a 20× fluorescence objective. Viral particles in conditioned supernatants (200 mL total volume) from 3 HBRV producing cell lines were pelleted by ultracentrifugation at 32,000 rpm for 1 h at 4 °C (SW 32Ti). The pellet was fixed for 10 min in 1% formalin at 4 °C with negative staining and processed for transmission electron microscopy (Hitachi H-7650 Transmission Electron Microscope).

Proviral HBRV was PCR-cloned and sequenced from total DNA extracted from HBRV positive Hs578T cells using betaretrovirus primers, as described [Accession numbers: JX843701, JX843702, JX843703, JX843704, JX843705, JX843706, JX843707, JX843708, JX843709] [20].

Ligation mediated (LM)-PCR was performed on genomic DNA extracted from HBRV producing Hs578T cells [19]. Briefly, genomic DNA was sheared with the Covaris shearer and double-stranded DNA adaptors were ligated onto the DNA (Linker-1 [GTAATACGACTCACTATAGGGCTCCGCTTAAGGGAC], Linker-2 [PO4-TAGTCCCTTAAGCGGAG-NH2]). The ligation products were then amplified using the HBRV long terminal repeat (LTR) Outer Primer (CGTCTCCGCTCGTCA CTTAT) and the Linker Outer Primer (GTAATACGACTCACTATAGGGC). Nested PCR was then performed with the HBRV-LTR Inner Primer (GCAGACCCCGGTGACCCTCAG) and the Linker Inner Primer (AGGGCTCCGCTTAAGGGAC) [19]. LM-PCR products were cloned into Illumina libraries using the Genomic DNA Sample Preparation Kit for paired-end next generation sequencing with the HiSeq 2000 or the MiSeq platforms.

### 2.3. Informatic Analysis of Integration Sites

The HBRV integration pipeline for Illumina next generation sequencing (NGS) libraries has been described in detail elsewhere [19]. Briefly, integrations were verified using the following criteria: (i) presence of the HBRV 3′ LTR fragment, (ii) a human sequence with at least 95% identity with the human genome (hg19 assembly) within three bases of the LTR [19]. The pipeline excluded proviral internal fragments primed from the LTR and false-primed human genomic sequences, as well as potential sources of contamination such as murine genomic DNA found in laboratory reagents [55] and any ambiguous sequence lacking 95% or more identity with the human genome. All HBRV integrations were mapped by sense or antisense direction to the closest genes (hg19 assembly) using Refseq gene definition (http://genome.ucsc.edu, accessed on 7 September 2021). The data for both HBRV in vivo and in vitro integrations were deposited into the NCI Retroviral Integration Database (https://rid.ncifcrf.gov/; accessed on 28 January 2021) [56]. To analyze the distance of insertions to transcription start sites (TSS), HBRV integrations were grouped in bins of 10 kb using a 200 kb window upstream and downstream proximal to genes. A random integration dataset was generated with a 1000-fold iteration within the human genome (hg19 assembly) and the median number of simulated integration sites with 95% confidence intervals was plotted within bins of 10 kb to the nearest TSS, as described [57]. A similar calculation was made for CpG islands without factoring in the orientation of the integration. Then, the statistical analyses were performed with Fisher’s exact test, using counts of categorical values of the number of experimental integrations in relation to the simulated experimental plots within specified regions proximal to the TSS and CpG.

Common integration sites (CIS) proximal to genes were assessed as previously described and defined as three or more independent HBRV proviral insertions arising in two different hosts within a specified range [52]. We calculated the genomic distance as an 88 kb window size for a probability of *p* = 0.01 for finding at least 2 more insertions to the right of a given insertion (Appendix A) using a combined database of both in vivo and in vitro HBRV integrations [58]. The CIS genes were analyzed to identify potential oncogenes using the following databases: the Candidate Cancer Gene Database [59], Tumour Associated Gene [60], Integrative OncoGenomics database [61], and the COSMIC Cancer Gene Census [62].

## 3. Results

### 3.1. HBRV Co-Culture Assay

To isolate the virus, we employed the same co-cultivation method used to demonstrate that PBC perihepatic lymph nodes harbor a transmissible agent with the capacity of triggering a disease specific phenotype in serial passage [21,48]. For these studies, lymph node homogenates from 4 PBC patients and 3 comparison samples were co-cultured with Hs578T cells, which are known to be permissive for MMTV replication [34]. Of the 7 lymph nodes chosen for study, 2 had been previously tested and found positive for HBRV proviral integrations [19]. Each of the 7 lymph node homogenates and a PBS control without homogenate were co-cultured in 6-well plates in triplicate with and without dexamethasone (Figure 1). One PBC and one control lymph node co-culture grew to 70% confluency after 7 days and each were tested for HBRV. All the supernatants were negative for RT activity by Cavidi, whereas the PBC lymph node homogenate was RT positive, and the control homogenate and PBS control were RT negative. We had anticipated difficulty in detecting the low viral burden of HBRV because the mouse betaretrovirus, MMTV, exhibits poor replicative activity in vitro when compared with other retroviruses [34,35,36]. Accordingly, we decided to perform repeated subcloning to derive virus (Figure 1).

**Figure 1 viruses-14-00886-f001:**
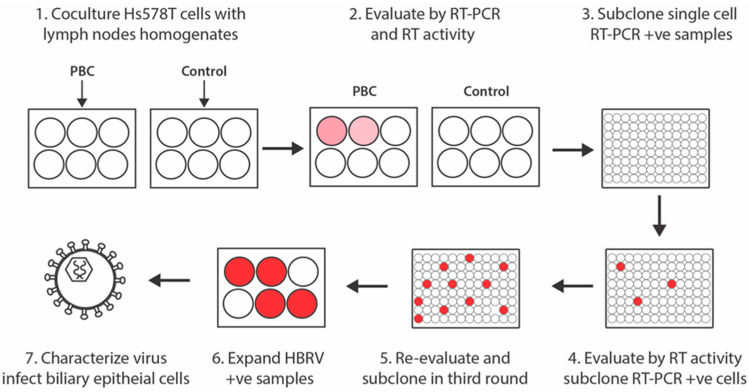
Isolation of HBRV by repeat subcloning of Hs578T cells infected with PBC lymph node homogenate. Co-cultures growing to 70% confluency with PBC lymph node homogenates were maintained in culture for 7 days. Then, three rounds of subcloning by limiting dilution of single cells exposed to PBC conditioned media, supernatants were assayed for RT activity and HBRV RT-PCR positive cells were subcloned for a total of three rounds [Pink and red circles represent increasing HBRV load with subcloning infected Hs578T cells].

### 3.2. Characterization of HBRV

Previous in situ hybridization studies with betaretrovirus *gag-pol-env* probes showed that 75% of biliary epithelial cells isolated from patients with PBC exhibited HBRV RNA signal, which was substantiated by the identification of proviral integrations by LM-PCR, and HBRV RNA using the Quantigene hybridization assay [19]. In the current in situ hybridization studies, we observed punctate HBRV RNA signal in the cytoplasm of Hs578T cells derived from the third round of subcloning (Figure 2B). To provide evidence of infectious virus, supernatants from infected Hs578T were pooled and co-cultured with uninfected biliary epithelium extracted from liver transplant recipients without biliary disease. HBRV RNA signal became detectable 8 days after co-culture and an increased percentage of cells exhibited signal by 12 days (Figure 2C), consistent with our prior experience of biliary epithelial cell infection [21]. Electron microscopy studies of pooled supernatants identified virus particles with membrane spikes and acentric cores (Figure 3) [21].

### 3.3. Cloning of HBRV Proviral Genome

The proviral genome was PCR-cloned from the infected Hs578T lines using oligonucleotide primers complementary to HBRV and then sequenced. The BLASTn searches established that individual clones shared 99% nucleotide identity with known HBRV and MMTV genomic sequences [2,3,12,18,20]. The confirmation of near identity between the human and murine betaretrovirus genomes emphasizes the likelihood of zoonotic transmission between mice and humans [63]. However, variants in the HBRV genome are detectable that translate to amino acid sequences not found in the MMTV envelope protein (Appendix A), for example, that signal unique features of the human agent. Recent studies suggest that the betaretrovirus genomic stability may be attributable to the enhanced activity of the MMTV polymerase, which has been shown to limit the mutability of the betaretrovirus genome [64]. While the limited variance observed in human and mouse isolates is a reproducible finding, it also continues to be a cause for concern with murine DNA contamination in PCR studies [16,41,65]. Nevertheless, our previous demonstration of more than 1500 HBRV proviral integrations in the human genome cannot be attributable to murine DNA contamination [19].

### 3.4. Evaluation of HBRV Proviral Integration Sites

#### 3.4.1. HBRV Integration: Transcription Start Sites (TSS) and CpG Islands

In prior in vivo and ex vivo studies using LM-PCR and NGS Illumina sequencing, we derived 1619 proviral integrations from liver, lymph node and biliary epithelium extracted from liver transplant recipients [19]. To prevent mis-primed human endogenous retrovirus (HERV) sequences being interpreted as HBRV insertions, the pipeline only recorded LM-PCR products with a 90% or greater identity with the reference betaretrovirus LTR as integrations. Even accounting for the potential of sequencing errors, this ensured a less than 1.4 × 10^−7^ chance that HERV LTR sequences would be accepted by the pipeline [19].

Furthermore, all potential contamination of MMTV integrations into the murine genome were sought and eliminated [19]. 

In the present study using the same pipeline and HBRV infected Hs578T samples, we derived a further 3408 proviral insertions (Figure 4A). The human Hs578T cells have not been passaged through mice nor do they have PCR evidence of MMTV-like DNA [34,36,49,50,51] and, therefore, these in vitro integration studies confirm the isolation of a human betaretrovirus derived from PBC patient lymph node samples.

The in vitro HBRV integrations were observed to cluster with a variable density of insertions within the human genome (Figure 4A). Accordingly, we assessed the relationship of HBRV insertions proximal to genomic features by comparing our experimental data with a plot of random insertion sites within the human genome. Generally speaking, individual retroviruses tend to adopt different behaviours and cluster around specific chromatin markers. However, MMTV appears to be an exception because the integration patterns show little demonstrable preference for insertions around TSS or CpG islands. For example, gammaretroviruses tend to integrate within +/- 2 Kb TSS and CpG islands through interactions of the viral integrase with bromodomain and extraterminal proteins; whereas human immunodeficiency virus has a propensity for avoiding CpG islands while preferentially inserting into transcription units guided by the LEDGF p75 transcriptional cofactor, which directly binds the HIV preintegration complex and tethers it to open chromatin [36,66,67,68,69].

To evaluate the proximity of insertions to TSS and CpG islands, we plotted the distribution of all the in vivo (see [19]) and in vitro HBRV integrations across the human genome (Figure 4A, HBRV in vitro). HBRV integration data were then compared to a dataset of random insertions into the human genome. For the in vitro analyses of integrations in Hs578T cells, analyses of integrations proximal to TSS showed no significant differences (Figure 4B). A slightly decreased frequency of HBRV insertions was observed in vitro 10kb proximal to TSS (experimental vs. simulated, 10.0% vs. 11.7%, *p* = 0.105) and a slight increase in frequency of in vivo integrations was found within the 10 kb approaching the 5′ of the TSS (experimental vs. simulated, 8.6% vs. 6.5%, *p* = 0.080). Accordingly, HBRV displays a random dispersion of integration sites proximal to TSS, similar to MMTV [36].

For the in vitro analyses of integrations in Hs578T cells around CpG islands, a diminished frequency of HBRV insertions vs. simulated was observed (25.6% vs. 32.0%, *p* < 0.001), whereas the opposite was found in the in vivo dataset with increased insertions around CpG islands (37.1% vs. 30.6%, *p* < 0.001). As the in vivo samples were derived from benign tissues and the Hs578T cells are derived from a breast cancer line, the differences observed in the two datasets may be in part attributable to hypermethylation of CpG islands in breast cancer [70]. To contextualize the findings of HBRV insertion in vivo, the difference in frequency of integrations was a small fraction of what may be seen with murine leukemia virus where insertions tend to be greater than five-fold increased within the 1–2 kb window proximal of CpG islands [67]. 

Notably, analyses of the MMTV integration site selection in Hs578T cells from our study showed decreased propensity to integrate close to either TSS or CpG islands (Figure 4D) as previously reported [36].

#### 3.4.2. HBRV Common Insertion Sites (CIS) Genes

MMTV was one of the first retroviruses shown to influence transcription of host genes by insertional mutagenesis and activation of proto-oncogenic pathways. Retroviral insertion mutagenesis screens often uncover integrations that upregulate transcription of multiple cellular genes over long distances following upstream insertion in the antisense orientation or downstream in the sense orientation of the gene [68]. Accordingly, we investigated the presence of common insertion sites (CIS) proximal to genes that may be transactivated as a result of HBRV integration. The CIS were defined as 3 or more independent HBRV proviral insertions arising in separate individuals (or experiments in vitro) found to be within an 88 kb range (Appendix A), which resulted in the identification of 44 genes. A database search was then performed to determine whether these 44 genes may be associated with carcinogenesis; 17 of the 44 genes analyzed were found to have a known association with at least once cancer disease site in at least one genetic database (Table 1). Of the 17 identified genes, 7 are involved in hepatocellular carcinoma (HCC).

Several HBRV CIS are in the vicinity of tumor-suppressor genes, including PTEN, RANBP2, ORC1 (Table 1). The loss of the tumor suppressor could contribute to a cancer phenotype. PTEN HBRV CIS, identified in both in vivo and in vitro samples, are located within the first intron in a plus orientation, and may lead to aberrant transcript generation. The loss of normal PTEN transcription is frequently associated with tumorigenesis with reduced expression leading to an epithelial-to-mesenchymal transition (EMT) and metastases. PTEN expression can be reduced via direct mutation, microRNAs, genome and epigenome changes and the reduced expression is implicated in many cancers, including HCC [71], breast cancer [72] and lymphoma [73].

#### 3.4.3. HBRV In Vitro Genome Clusters

While HBRV displayed limited bias when integrating into the human genome in vitro (Figure 4A), nine chromosomal regions demonstrated clustering of independent, mainly intergenic, integration sites, in both cis-plus and, -minus, orientations. In six of the nine regions, genes located upstream and downstream of an integration are implicated in liver and/or breast cancer (Appendix A). In HCC, SPINK1 (5q32) overexpression supports cell proliferation and metastases [74]. ZFPM2-AS1 (8q22.3) and Mir30B (8q24.22) are implicated in both HCC and breast cancer. In HCC, ZFPM2-AS1 interacts with miR-576-3p to up-regulate HIF-1α [75]. In both HCC and breast cancer, high level expression of ZFPM2-AS1 is associated with advanced disease and cell proliferation [75,76]. The mir30 gene family perform tumor suppressor functions, and mir30B limits lung invasion by liver tumors [77], and may limit bone invasion by ER-/PR-negative breast cancer cells [78]. The HBRV integrations identified occur at a distance from the genes described, but are in orientations that could alter the expression of genes that influence HCC and breast cancer progression (Appendix A) [79].

## 4. Discussion

In this study, we isolated infectious virus particles with genomic and morphological resemblance to HBRV using Hs578T cells co-cultured with PBC patient samples. The infectious lymph node homogenates used for the co-culture studies were derived from samples that had previously been shown to be positive for HBRV proviral integrations or HBRV RNA by RT-PCR. We chose to infect the Hs578T breast cancer cells as they are permissive for MMTV infection. As expected, a low level of viral replication was found, requiring subcloning virus-producing cells to reach a sufficient level of production exceeding 1000 genome equivalents per cell (Appendix A). We were then able to isolate virus from supernatants from infected Hs578T cells and passage infection to primary cholangiocytes. Subsequently, we investigated the presence of proviral integrations in vitro to provide robust evidence of HBRV infection. We performed these gold-standard studies as prior critiques have suggested that PCR studies in patients with breast cancer may reflect contamination of mouse DNA [16,40,41]. In this study, we identified over 3400 HBRV integrations in vitro to confirm the presence of HBRV.

Our studies were first conceived to isolate HBRV and demonstrate transmissibility, in part to address Koch’s postulates in vitro. When we first characterized HBRV in PBC patients, we found that the perihepatic lymph nodes were the predominant reservoir of infection that contained infected monocytes displaying HBRV capsid and surface proteins. Notably, the same infected cells had markedly increased and aberrant expression of autoantigens reactive to the diagnostic anti-mitochondrial antibodies [21]. As this phenotype is highly specific for PBC patients and found in damaged bile ducts, the concurrent detection of pathology and viral proteins in the same cells strongly suggested that infection may be altering mitochondrial protein expression.

To test the hypothesis, we first co-cultivated PBC lymph node homogenates with healthy cholangiocytes and this led to aberrant expression of mitochondrial antigens in cholangiocytes, which only occurred with use of the PBC conditioned co-cultures [21,48]. Serial passage of PBC conditioned media also triggered the mitochondrial autoantigen expression, which could be abrogated with gamma irradiation of lymph node homogenates. We then characterized the particles as bearing the hydrodynamic properties of retroviruses, containing the HBRV genome and having the electron microscopy appearance of B-type particles with acentric nuclear core (Figure 3). Subsequently, we showed that the mitochondrial changes could be triggered in primary cholangiocytes by infecting them in co-culture with pure MMTV isolates [21]. Subsequent studies have revealed that the metabolic remodeling in PBC biliary epithelium involves HIF-1α pathway activation, predominant use of glycolysis versus oxidative phosphorylation, and mitochondrial inhibition resulting is a compensatory mitochondrial biogenesis [23]. Similar metabolic changes were described with MMTV infection and mitochondrial biogenesis in mouse breast cancer [84]. 

MMTV was the first oncogenic mammalian retrovirus shown to induce cancer by insertional mutagenesis leading to increased expression of cellular proto-oncogenes [85]. The functional insertions are usually found upstream of genes in the antisense orientation or downstream in the sense orientation to prevent the positioning of the viral promoter between the enhancer in the 5′ LTR and the host gene [79]. The enhancers may augment the activity of promoters over large distances via chromatin loop interactions permitting upregulation of multiple genes. Notably, the first characterized CIS for MMTV in mouse breast cancer included members of the *Wnt*, *Notch* and *FGF* developmental pathways, some of which were unexpectedly found to be activated in a microarray study of liver samples derived from patients with PBC [85,86]. Similarly, studies comparing MMTV insertions and transcriptional dysregulation leading to breast cancer in mice are directly comparable to transcriptional changes observed in patients with breast cancer [52]. As we derived over 5000 HBRV integrations in vivo and in vitro, we looked for evidence of HBRV common insertion sites to assess the potential relationship with cancer genes, analogous to site preference employed for gammaretroviruses FeLV in MCF-7 human tumor cells [68] and CIS for MMTV [52].

PBC patients in general are at increased risk of cancer, which may be in part related to their diminished immunity [23]. A multi-national metanalysis of over 16,000 PBC patients reported an increased relative risk of hepatocellular carcinoma (HCC), breast cancer, lymphoma, renal cancer and other female hormone responsive cancers [87]. While PBC patients are at increased risk of HCC, it is notable that all patients with cirrhosis have increased risk of HCC as well [87,88,89]. Therefore, it is not possible to infer that HBRV may be linked with HCC or breast cancer based on the epidemiological data alone. However, patients with chronic viral hepatitis, including those with either hepatitis B or hepatitis C virus infection, usually have a far higher risk of HCC as compared to non-viral causes of cirrhosis [89].

The accumulated data suggest that HBRV is probably a hepatotropic virus [19,22,23]. In the nested PCR survey, HBRV was detected in 19% of HCC tumor tissues and not in healthy liver samples; HBRV DNA was found in the liver of 12% of PBC patients (concordant with our data), 25% of patients with other hepatic disorders (including those with AIH with a higher frequency of HBRV [19]) and up to 47% in those with alcohol use disorder [22]. Collectively, these studies begin to contribute to growing body of evidence that HBRV is a hepatotropic virus that is linked with different diseases depending on genetic predisposition (PBC and AIH), or comorbid conditions (alcohol use) [22,23,28]. As discussed, we still lack robust and reproducible diagnostic assays to record the true prevalence of HBRV in patients with liver disease.

As HBRV is found at a higher frequency in patients with HCC and is an oncogenic virus, we sought evidence that it may preferentially integrate in close proximity to HCC-related genes. Our observation that 15% of the non-random, cancer-associated HBRV CIS are proximal to genes implicated in HCC supports the potential role for HBRV in human cancer, including HCC. Two well-known tumor suppressors, PTEN and RANBP2, were HBRV targets, indicating that loss of gene function, e.g., via insertional mutagenesis or aberrant splicing, might be the mode of action required to silence tumor suppressors, as opposed to the MMTV-enhancer action at proto-oncogenes [52,85]. Loss of tumor-suppressor function has been observed for FLV retroviral integration at p53 [79,90].

There are aspects of the study that could have been improved. The use of Hs578T cells to isolate HBRV was both a strength and weakness of the study as the cells are permissive for MMTV and may therefore be susceptible to contamination from outside sources. However, no such MMTV-producing cells were used within the laboratory at the time of the co-culture studies, and all co-cultures were performed in a cell culture room with human samples only. Another potential weakness of the study was not identifying viral proteins in infected cells rather than proviral integrations and DNA FISH hybridization could have been employed to quantify proviral integrations per cell. Differences were observed with HBRV in vivo and in vitro data, which may alter integration patterns. This in part may be related to methylation at CpG islands in the Hs578T cells but also because the Hs578T karyotype may explain some of the large-scale differences seen in the ideogram (Figure 4A). Accordingly, the CIS data serve to provide preliminary data for future integration studies in other diseases and tissue types.

## 5. Conclusions

An exogenous and transmissible human betaretrovirus closely resembling MMTV has been isolated from patients with the autoimmune liver disease, PBC. Preliminary studies characterizing common insertion sites in liver disease patients and the Hs578T breast cancer line in vitro demonstrate integrations proximal to cancer related genes associated with hepatocellular carcinoma and breast cancer. Given that patients with PBC respond to antiviral therapy and that the global incidence of breast cancer continues to rise, these findings suggest that the role of HBRV in human disease merits further investigation. 

## Figures and Tables

**Figure 2 viruses-14-00886-f002:**
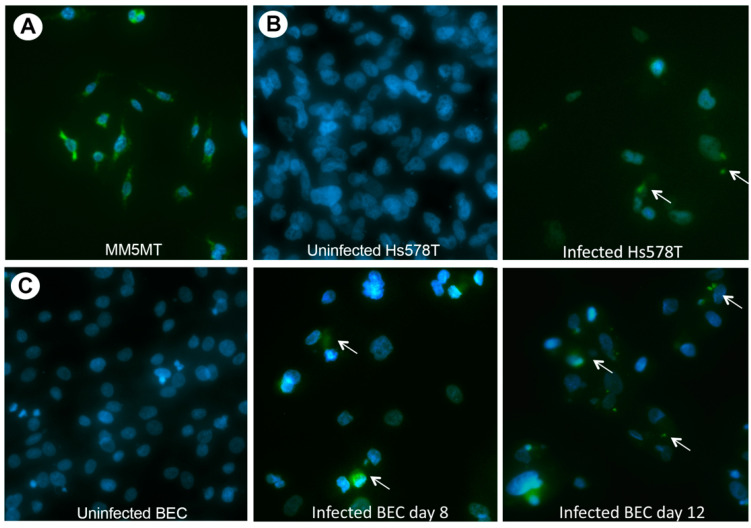
Dark field microscopy showing betaretrovirus RNA by in situ hybridization (green signal) (**A**) MMTV RNA in the MM5MT murine breast cancer cells (control), (**B**) HBRV in the cytoplasm of Hs578T following co-culture with PBC conditioned media and (**C**) HBRV signal in biliary epithelial cells infected with lymph node conditioned Hs578T supernatants [white arrows indicative of HBRV RNA, DAPI blue stain for nuclei, magnification ×200].

**Figure 3 viruses-14-00886-f003:**
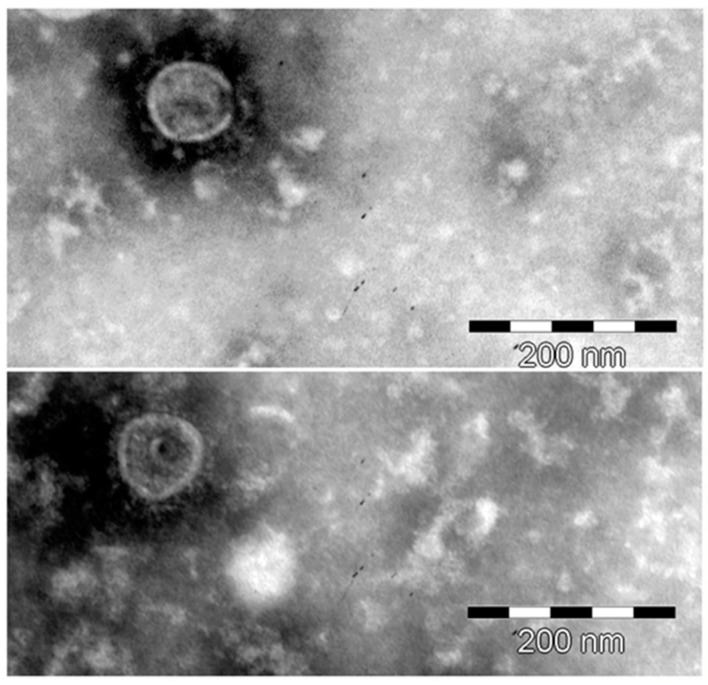
Negatively stained supernatant from infected Hs578T cells was analyzed by transmission electron microscopy to identify 85–95 nm virus-like particles with acentric cores and membrane spikes, comparable with MMTV and prior betaretrovirus-like particles derived from patients with PBC.

**Figure 4 viruses-14-00886-f004:**
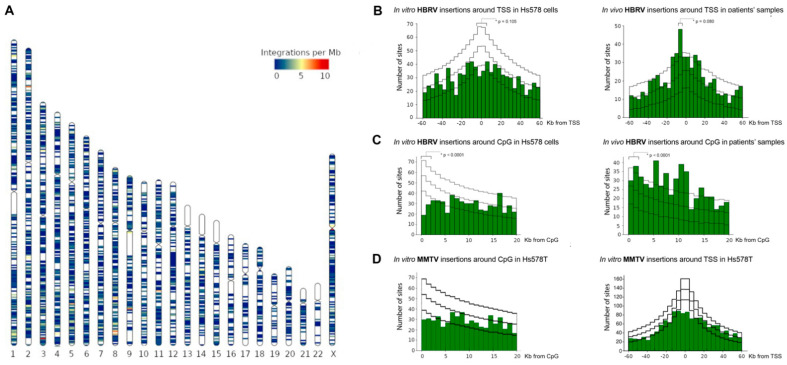
(**A**) HBRV integrations in vitro demonstrate a variability in the density of sites with areas of clustering within the genome [with red/orange signal demonstrating 5–10 frequency of integrations per Mb on Chr 3q, Chr 5q, Chr 8q, Chr 9q, Chr 11q, and the centromere on X Chr.] (**B**) Compared to a computer-generated set of random integrations, HBRV showed little difference from the random plot of insertions within the genome with regard to integration proximal to TSS. (**C**) HBRV avoided insertion within 2 kb proximal to CpG islands in the Hs578T breast cancer cell line but aggregated proximal to CpG in the samples derived from patients with liver disease. (**D**) In the Hs578T breast cancer cell line, MMTV integrations did not aggregate around CpG and TSS as previously reported [36]. [The three black lines represent the median and range [5–95%] of insertions randomly generated with 1000 iterations throughout the human genome].

**Table 1 viruses-14-00886-t001:** HBRV integration sites: common insertion site (CIS) genes associated with human tumors.

Gene Symbol	Integration Orientation	Candidate Cancer Gene Database	Tumor Associated Gene	Integrative Oncogenomics	COSMIC Cancer Gene Census	Relevant Cancer Type
PTEN *	(+)	Y **	Y	Y	Y-Tier 1/Hallmark	Liver [71], Breast [72], Lymphoma [73]
RANBP2 *	(+)	NL		Y	Y-Tier 1/Hallmark	Liver [80], Breast [81]
PRAMEF8	(+)	Y				
MB21D2 *	(+)	Y		Y	Y-Tier 2	Liver [82]
CLEC2L	(+)	Y				
KLRG2	(+)	Y				
ZFAT	(+)	Y				
MAGEB5	(+)	Y				
HEATR1	(+)/(-)	Y				
ORC1 *	(+)/(-)		Y			Breast [83]
FARP1	(+)/(-)	Y				
STK24	(+)/(-)	Y				
BRMS1L	(+)/(-)	Y				
DCAKD *	(+)/(-)	Y				
SSB *	(+)/(-)	Y				
SPOCK3 *	(+)/(-)	Y				
FAM73B	(+)/(-)	Y				

* Genes known to be involved in hepatocellular carcinoma. ** Y—well-established association with cancer.

## Data Availability

The datasets used and/or analysed during the current study are available from the corresponding author on reasonable request.

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
