# Peer review of "Isolation of a Human Betaretrovirus from Patients with Primary Biliary Cholangitis"

_viruses, 2022, doi:10.3390/v14050886_

Round 1
Reviewer 1 Report
I read with great interest the research article by Goubran et al entitled "Isolation of a Human Betaretrovirus from Patients with Primary Biliary Cholangitis". The authors describe efficiently and in a very analytical manner the potential role of Human Betaretrovirus. They managed to isolate human betaretrovirus from patients with PBC and also they showed that the mitochondrial changes could be triggered in primary cholangiocytes by infecting them in co-culture with pure MMTV isolates, providing evidence for pathogenetic mechanisms. They also provide evidence of how human betaretrovirus could be involved in carcinogenesis.
It would be very interesting to see also what happening in other diseases such as chronic hepatitis C, chronic hepatitis B, autoimmune hepatitis, NASH. This is a limitation of the study, which it should be mentioned.
Author Response
Reviewer 1: It would be very interesting to see also what happening in other diseases such as chronic hepatitis C, chronic hepatitis B, autoimmune hepatitis, NASH.
We thank reviewer 1 for their helpful review and comments. We have added the relevant data from two prevalence studies. We have discussed the prevalence of HBRV in patients with PBC and the related liver disease autoimmune hepatitis (AIH) in the introduction (lines 87-103). We also amended the discussion to discuss the prevalence of HBRV in patients without PBC in more detail (lines 462-472). The discussion is limited somewhat as we still lack robust and reproducible diagnostic assays to record the true prevalence of HBRV in patients with liver disease.
Reviewer 2 Report
The laboratory of Prof. Mason has been a leader in studying the linkage between HBRV and PBC for the last 20 years. In the present manuscript they test the hypothesis that HBRV from lymph nodes of PBC patients is transmitted and (integrates) into a breast cancer cell line as well as into biliary epithelial cells that lack HBRV. This is tested via co-culturing followed by different analyses. One shortcoming of the manuscript is the lack of viral protein identification in the infected cells (as mentioned by the authors themselves).
According to previous publications by the authors, they have anti-MMTV antibodies and have already shown positive immune histochemistry in PBC HBRV positive FFPE sections of lymph nodes. So it should be very easy to add a slide of the biliary epithelial cells infected by the virus using the available antibodies. This would add to the appeal of the information if antibodies are available now.
Still the manuscript in its present form is acceptable
Some required clarifications (in the discussion)
- What is the percentage of PBC HBRV positive patients with regards to HBRV negative PBC patients
- Do PBC HBRV positive patients have an increased tendency for developing breast cancer
- And by the same token, do women with HBRV (HMTV) associated breast cancer demonstrate enhanced tendency of developing HBRV associated PBC.
- Based on the detailed integration analysis carried out here, are there any candidate signaling pathways that are more likely to be associated with the disease.
Author Response
Reviewer 2: According to previous publications by the authors, they have anti-MMTV antibodies and have already shown positive immune histochemistry in PBC HBRV positive FFPE sections of lymph nodes. So it should be very easy to add a slide of the biliary epithelial cells infected by the virus using the available antibodies. This would add to the appeal of the information if antibodies are available now. Still the manuscript in its present form is acceptable
We thank reviewer 2 for their helpful review and comments, some of which overlap with reviewer 1. We agree that we would have liked to have performed the suggested immunochemistry studies have been unable to do so due to sample, reagent, manpower and other issues related to the pandemic.
What is the percentage of PBC HBRV positive patients with regards to HBRV negative PBC patients
We have summarized the percentage of PBC HBRV positive and negative PBC patients in the introduction and discussion (lines 87-103 and lines 462-472). In summary, the majority of PBC patients have infection.
Do PBC HBRV positive patients have an increased tendency for developing breast cancer. And by the same token, do women with HBRV (HMTV) associated breast cancer demonstrate enhanced tendency of developing HBRV associated PBC.
I personally see patients developing PBC following chemotherapy treatment for breast cancer, but this is anecdotal. However, we do not know whether PBC patients with HBRV have an increased tendency for developing breast cancer or whether women with HBRV associated breast cancer demonstrate a higher risk of developing PBC (line 453). Further epidemiological studies are required to address this specific question.
Based on the detailed integration analysis carried out here, are there any candidate signaling pathways that are more likely to be associated with the disease?
Based on our integration analysis, we are unable to directly implicate candidate signaling pathways that are more likely to be associated with PBC. Notably, the first characterized CIS for MMTV in mouse breast cancer included members of the Wnt, Notch and FGF developmental pathways, some of which were unexpectedly found to be activated in a microarray study of liver samples derived from patients with PBC (see lines 437-440).